# Understanding and Balancing Generalist-Specialist Approaches in Dementia Research and Care Practice, Qualitative Research with 44 Dementia Professors in The Netherlands

**DOI:** 10.3390/ijerph20053835

**Published:** 2023-02-21

**Authors:** Robbert Huijsman

**Affiliations:** Erasmus School of Health Policy & Management, Erasmus University Rotterdam, P.O. Box 1738, 3000 DR Rotterdam, The Netherlands; huijsman@eshpm.eur.nl

**Keywords:** dementia, generalist, specialist, academic professors, qualitative research

## Abstract

Dementia is one of the leading causes of death and disability among citizens and a societal challenge because of aging worldwide. As dementia has physical, psychological, social, material, and economic impacts, both research and care practice require many disciplines to develop and implement diagnostics, medical and psychosocial interventions, and support, crossing all domains of housing, public services, care, and cure. Notwithstanding large research efforts, much knowledge about mechanisms, interventions, and needs’ based care pathways is still lacking. To cope with these challenges in research and practice, this paper is the first to question how generalist and specialist orientations can be unfolded. In the Netherlands, all dementia professors (N = 44) at eight Dutch academic centers have been interviewed. Qualitative analyses revealed three subgroups of dementia professors, one with a generalist orientation, one adhering to specialist approaches, and a third group that pleas for mixed orientations, with some differences between research and care practice. Each group has arguments for its generalist/specialist vision, but the synthesis suggests a paradigm of personalized and integrated dementia care, aimed at the individual in his own living environment. Sustainable strategies to cope with dementia require (inter)national programs and strong collaboration to build multi- and interdisciplinarity within and between research and practice.

## 1. Introduction

Dementia is a clinical syndrome of impairment in multiple cognitive domains (e.g., memory, language, an executive function that includes self-regulation, task planning, and execution) affecting activities of daily living and everyday function that is progressive in nature and not the result of reversible causes, resulting in uncertain clinical trajectories [1]. Alzheimer’s disease is the most common form of dementia (60–70% of cases). Other major forms include vascular dementia, dementia with Lewy bodies (abnormal aggregates of protein that develop inside nerve cells), and a group of diseases that contribute to frontotemporal dementia (degeneration of the frontal lobe of the brain). Dementia may also develop after a stroke or in the context of certain infections such as HIV, harmful use of alcohol, repetitive physical injuries to the brain (known as chronic traumatic encephalopathy), or nutritional deficiencies. The boundaries between different forms of dementia are indistinct and mixed forms often co-exist.

The WHO estimates that more than 55 million people (8.1% of women and 5.4% of men over 65 years) are living with dementia. This number is estimated to rise to 78 million by 2030 and to 139 million by 2050 [1]. Dementia does not exclusively affect older people—young onset dementia (defined as the onset of symptoms before the age of 65 years) accounts for 5–10% of all persons with dementia [2]. Dementia is currently the seventh leading cause of death among all diseases and one of the major causes of disability and dependency among older people worldwide. According to the EU Joint Programma Neurodegenerative Disease Research (JPND), dementia is responsible for already the greatest burden of disease, affecting over 7 million people in Europe and this figure is expected to double every 20 years as the population ages [3]. It currently costs approximately €130 billion per annum to care for people with dementia across Europe, highlighting age-related neurodegenerative disease as one of the leading medical and societal challenges. Alzheimer’s disease is particularly expensive to manage due to its insidious onset, its ever-increasing levels of disability, and the length of time over which the condition extends itself. The average duration of this disease is between two and 10 years, during which patients will require special care that is a significant burden for both caregivers and for society as a whole.

Dementia has physical, psychological, social, material, and economic impacts, not only for people living with dementia but also for their carers, families, and society at large [1]. Especially in the later stages of the disease, care required for people with dementia includes primary health care, community-based services, specialist care, rehabilitation, long-term care, and palliative care. The behavioral symptoms of dementia require an interdisciplinary approach in both practice and research [4]. Dementia requires integrated care, i.e., a coordinated way of working across multiple professionals, organizations, and sectors in order to improve the health, quality of care, and economic outcomes for a targeted (sub)population [5]. The type of integration refers to four domains: (1) services integration, (2) professional integration, (3) organizational integration, and (4) systems integration of policy and regulation [5]. However, political interests and national health systems determine whether dementia care has a medical, social care, or integrated health and social care focus [6]. The immaturity of general and specialized services provided by the health and social care sectors also determines the level of informal care, which is primarily provided by family members. 

The WHO declared dementia a public health priority [7]. In May 2017, the World Health Assembly endorsed the Global action plan on the public health response to dementia 2017–2025 [8]. This action plan provides a comprehensive blueprint for action—for policy-makers, international, regional, and national partners, and WHO in the following areas: addressing dementia as a public health priority; increasing awareness of dementia and creating a dementia-inclusive society; reducing the risk of dementia; diagnosis, treatment and care; information systems for dementia; support for dementia carers; and, research and innovation. Special attention is also required for end-of-life, i.e., palliative care. The WHO has developed A Blueprint for Dementia Research [9], together with researchers and academics around the world, i.e., a global coordination mechanism to harmonize the global dementia research and innovation agenda, synergize efforts, and stimulate new initiatives. Moreover, WHO urged all countries to develop a National Dementia Strategy or Plan, i.e., a written organized set of principles, objectives, and/or actions for reducing the burden attributable to dementia in a population. A dementia plan may be stand-alone, i.e., specific to dementia, or integrated into other mental health, aging, non-communicable diseases, or disability [10]. 

In the Netherlands, the National Dementia Strategy of the Ministry of Health, Welfare, and Sport is intended to function as a driving force that will ensure dementia research continues and accelerates over the next ten years (2021–2030). Over 140 million euros will be invested in research, with a focus on large multidisciplinary research consortia [11]. These multidisciplinary consortia will bring together existing top-ranking centers with other research groups and disciplines, healthcare facilities, teaching institutions, and representatives of the business community and society at large. The role of the consortia is to ensure the integration of new knowledge about dementia in teaching, research, and healthcare. Long-term funding of these research consortia is aimed to create a knowledge, research, and development infrastructure that is broad-based and long-lasting. Finding solutions for people with dementia now and in the future demands an approach in which research, development, improvements to practice and social innovation go hand in hand. 

Optimizing the roles played by both generalists and specialists in (research into) the diagnosis and treatment of dementia could have a major impact on the quality and cost of patient care. Yet, research on the value of generalist and specialist approaches is scarce, both in the field of daily care practice as in the world of interdisciplinary research. There is evidence in the literature suggesting differences between specialists and generalists in terms of knowledge, patterns of care, use of resources, and clinical outcomes of care for a broad range of diseases [12]. According to Pearson [13], generalists and specialists have always faced conflicts in competing for patient loyalties, professional prestige, and compensation, and managed care and disease management have fueled these tensions even more. The IMPACT study by Robinson et al. is one of the few to survey 250 generalists and 250 specialist physicians in dementia care, from five different European countries, and found more similarities than differences between specialists and generalists regarding a broad spectrum of issues relating to Alzheimer’s; differences between countries appear to be greater than differences between physician groups [14]. The major difference between the two types of physicians was in terms of treatment initiation; twice as many specialists as generalists reported initiating therapy and specialists are generally more knowledgeable about the management of chronic disease and about their area of expertise, earlier apply more intensive therapies and quickly adopt new and evidence-based treatments than generalists [12,14]. Generalists might have a “wait and see” attitude, possibly from a greater concern about therapeutic complications, or greater caution in accepting new data or changing established patterns of treatment [12]. However, scarce studies, such as IMPACT, did not investigate underlying professional beliefs to clarify and characterize possible similarities and differences between generalist and specialist visions of dementia.

Dementia care networks bring together various primary and secondary health care providers, outpatient and inpatient services, community and private sector organizations, doctors, therapists, specialists, and volunteers [15]. Some dementia care networks have a more health-oriented approach and link services often related to the diagnosis, treatment, and care of a person with dementia, whereas other networks take a broader approach that focuses on person-centered coordinated care and support, often under the broader heading of “frail older people” instead of the disease-specific heading of “dementia”. One of the challenges is the ongoing discussion about (the balance between) disease-specific or more generalist approaches for dementia in research, teaching, and healthcare practice. This is for instance visible in the debate about case management dementia versus community nursing and about the continued existence of dementia networks or merging into broader partnerships for frail elderly. However, also in the social services by municipalities, which do not differentiate according to target groups or syndromes, and in the medical specialization line between general practitioner, internist, specialist geriatric medicine, and clinical geriatrician. In palliative care, an important part of the care continuum for patients with cancer, dementia, and other progressive chronic diseases, there is also an ongoing debate about specialist versus non-specialist and generic or disease-specific approaches, as demonstrated in a review article by Nevin et al. [16]. One of the underlying assumptions or philosophical views that palliative care is part of the remit and workload of every healthcare practitioner who is in contact with patients with chronic illnesses and not just the responsibility of specialist palliative care services also emerged as a precondition in the review of Nevin et al. [16]. In an earlier review of the interface of specialist and non-specialist palliative care, Gardiner et al. concluded that clear definitions of roles and responsibilities are required as a priority in order to address the professional territorialism that exists in palliative care provision [17].

To meet the complex societal health challenges, authorities, and research institutions emphasize and encourage interdisciplinarity in dementia research, i.e., communication and collaboration between researchers across academic disciplines [4]. Interdisciplinary research is based upon a conceptual model that links or integrates theoretical frameworks from those disciplines, uses study design and methodology that is not limited to any one field, and requires the use of perspectives and skills of the involved disciplines throughout multiple phases of the research process [18]. The differences in training, values, and experiences allow different disciplines to view health challenges from different angles. Together, this could provide a broader view and better knowledge than multidisciplinary research, but interdisciplinarity will only succeed if different scientific disciplines manage to integrate their theoretical frameworks [4,18]. An interdisciplinary approach benefits research in many ways: team members from different disciplines can share skills, expertise, knowledge, and experience throughout a project and develop a richer and more complex understanding through enhanced group reflexivity and triangulation of results [19]. Reeves et al. stressed that teamwork between professionals of different disciplines is just one of the forms of interprofessional work and presented four different types of interprofessional practice, ranging from light to intertwined: interprofessional networks, interprofessional coordination, interprofessional collaboration, and interprofessional teamwork [20]. 

This paper’s aim is to explore the arguments in the discussion about generalist versus disease-specific approaches to dementia in the Netherlands. For this, all Dutch academic professors dedicated to the field of dementia in the Netherlands have been interviewed. This topic was part of a broader interview on the developments and challenges in research and care in the dementia field in the Netherlands [21]. Dutch dementia professors were asked to reflect on the following question: “What do you consider best for integrated care and support for people with dementia, a disease-specific or a more generalist approach? Why is this, from your (research) experience”? As a fellow professor, my overall purpose was to investigate how all dementia professors could arrive at a shared vision about future directions for dementia research and dementia care and to develop sustainable solutions for all the demographic, financial, and societal challenges we face.

## 2. Methods 

### 2.1. Design

This qualitative research is based on semi-structured interviews with all dementia professors from all eight universities (not the technical universities at Delft, Eindhoven, or Twente/Enschede) and academic hospitals in the Netherlands. These dementia professors were included from all relevant disciplines and under the restriction that at least half of their academic work is devoted to dementia. The list of possible respondents stems from different sources, such as the website of ZonMw (a national research funding organization), Alzheimer Nederland, personal network (RH is himself a professor in elderly and dementia care since early 1997), and additional “snowballing” amongst professors themselves. During this recruitment process, the list became over-complete; seven invited professors declined participation because dementia was a minor part (10–15%) of their work.

### 2.2. Interviews

The author reached out to all dementia professors by email or phone call and performed all interviews for this research and the writing of this paper alone. The interviews took place in the period from August 2018 to March 2019. Prior to the interviews, each interviewee received an email containing information about the research objectives and an informed consent form seeking their permission to transcribe and cite the interview. All professors (N = 44) gave informed consent by email to the author and repeated that at the start of each interview (part of the transcription). There were no dropouts during the whole process. Ideally, the interviews would take place in their own working location, but due to full agendas and travel time, about two-fifths of the interviews were completed by telephone. On average, the interviews took about one hour, plus or minus 15 min. Often, the respondents shared additional publications of their own or their research group for further information. All information was also used to develop a short CV of each professor, including their most important research lines and 10 to 15 key publications with hyperlinks [21]. All interviews were recorded after approval and transcribed by experienced professionals (marinka^®^transcriberen, website https://www.transcriberen.nl/, accessed on 20 February 2023). Transcripts were sent back to the respondents (“member check”), for possible corrections or additional comments and to reconfirm informed consent and the usage of quotes, after anonymization and with a little editing for the purpose of readability. Only a few respondents added their comments, which are included in the results. 

### 2.3. Data Analysis

To provide a rich description of the sample of all Dutch dementia professors, SPSS (version 28.0.1.0) was used to test for possible differences in socio-demographics, disciplines, and settings of the respondents (see Section 2.4).

All transcripts were entered in the qualitative data analysis program ATLAS.ti (version 9). The qualitative analysis consisted of steps inspired by the grounded theory, originally developed by Glaser and Strauss [22]. The interviews’ transcripts were analyzed using a three-step coding process involving open, axial, and selective coding [23]. After filtering the themes, quotes have been selected to give in-depth illustrations and explanations of the various opinions of the dementia professors. That will be completed in quite a few details to help the readers develop their own opinions in the generalist-specialized debate. Personal visions on generalist and/or specialist approaches in research and practice were asked and discussed directly in person with all respondents.

### 2.4. Respondents

Around the time of interviewing, there were 44 professors active in the field of dementia research, working at eight universities in the Netherlands (see Table 1). The gender distribution is not quite equal (43% female, 57% male) and the average age is 54.2 years (SD 9.5 years), with 16% below 45 years and 11% 65-plus. On average, male professors are 4.2 years older than female professors (56.0 versus 51.9), but this difference is not significant (*t* = 1.469; *p* = 0.075). Most respondents work in the academic institutions in Amsterdam (38%; a merger between Free University Amsterdam and Amsterdam Medical Center, established in June 2018, just before the start of the interviews), followed by Groningen (16%), then Maastricht and Nijmegen (each 11%), Rotterdam (9%) and then Utrecht and Leiden and Tilburg (all 5%). The last university is the only one without an academic hospital. The first five academic hospitals also run a (tertiary) Alzheimer’s Center for diagnostics and dementia care. Disciplinary backgrounds vary widely, in both the initial education and the current discipline in which the professor holds their chair.

At the moment of the interview, respondents hold their (first) chair for an average of 9.1 years (SD 7.9 years; no significant gender differences), with 30% shorter than five years and 7% longer than 20 years. One out of five now holds a second or even third position as a professor, starting on average four years after the first position. To become a first-time professor at an average of 45 years takes a long time of (academic) education (26 years), Ph.D. research (9 years), followed by further academic work and publishing (10 years) (see Figure 1). Again, there are no significant gender differences, nor in the age of finishing a Ph.D. (average age for males 34.63 years and females 34.56 years; *t* = 0.043, *p* = 0.483), nor in the age of becoming a professor (males at an average age of 45.6 and females 44.2 years; *t* = 0.725, *p* = 0.236).

## 3. Results 

### 3.1. Finding First Patterns in All Views of Dementia Professors

Should you choose a generalist approach, or should you focus on disease-specific (research on) interventions in specialized care fields? A first round of coding of the views of the dementia professors reveals that they react very differently and do not seem to want any credit for final wisdom. For example, a very experienced Groningen professor of geriatric medicine and dementia shares his doubts: 

“*I am struggling with that myself and that is of course also the political theme. Then it is about case management dementia, you know how sensitive that subject is. Then you can say: we are going to organize the entire care around a disease. Knowing that this disease also has a number of very specific characteristics in its course, so that you can also very well seize the opportunities to train people specifically. Or should you take the generalist approach to frail older people, knowing that people with dementia also have three or four other diseases. And that part of the strength of healthcare innovators lies in better collaboration, cross-connections, and better overall expertise. I find that a very difficult discussion*.” [G7]

During the interviews, three schools of thought emerged, each with some underlying sub-conceptions: a generalist approach, a generalist basis with specialist deepening or supplementation, and an entirely disease-specific approach. Then it also matters to some professors to make a distinction between research and practice, and many nuances are given: 

“*If you want to advance science, it makes sense to look at dementia separately, in research you want to have patient populations that are as clearly defined as possible. This is how the Alzheimer Center in Amsterdam works, which focuses mainly on people who develop dementia at a younger stage of life. And in which genetic components are also present. But if you look at it from the care aspect, I believe more in a broad approach in which you can easily involve your colleagues from the various disciplines. So that you can easily perform additional diagnostics in a multidisciplinary way, for example*.”[A15]

Table 2 gives an overview of how the professors position themselves in the debate about generalist versus specialist approaches and whether that differs between research and practice. That seems to be the case, as signaled by the Chi-square test (*p* < 0.001). A specialistic approach is more frequently preferred in care practice than in dementia research. 

“*A specific focus might be useful when it comes to research in the brain, but not when it comes to care. Because dementia is, in the first place, a fairly broad collective term, it is a syndrome. And secondly, there is a lot of co-morbidity and co-problems. So, if you only knew about dementia, you would be behind the facts. It is always the interplay*.”[A11]

A number of respondents emphasize that the views also depend on one’s position in the healthcare field. In primary care, it makes sense to think and act in a more generalist way, because the population there is more diverse and the nature of the demand is more general in the first instance. However, if you move further in the secondary and tertiary line, you can expect a more specialized approach, as one professor emphasizes this for nursing homes: 

“*For dementia as a total syndrome, a separate approach is absolutely necessary, because the disease affects you at the core of your existence. And the entire repertoire that you can normally use if you have problems dealing with those problems. The care providers in nursing homes are completely tuned to this.*
*But within the total syndrome, specific approaches to specific etiologies in care do not work, because the care has to look at what is bothering someone and how you can sustain or improve quality of life. That is much more important than any etiological mark on it.”*
[U1]

A primary care professional cannot be a specialist in everything but may have specialists in the primary group practice. However, then it becomes an issue to keep to the perspective of integrated care. From an organizational and efficiency perspective this is a very complex issue.

“*On the one hand dementia has something specific in terms of syndrome and that really requires knowledge of what to do and how to deal with it. But how to organize is a completely different question. Who does what and how do we do it in the region? That is a question of scale and efficiency. And then I would say that there are quite a few things that can be generic because there are also many similarities. Everyone says that it is very unique and in a way that is true, but if you look closely, the underlying mechanisms also have a lot in common*.”[T2]

In total, 31 out of the 44 respondents (70.5%) do not differentiate between practice and research (presented at the diagonal in Table 2). In the generalist-specialist debate, there are no significant differences between ages, disciplines, or universities of the dementia professors, although for research, Amsterdam is more on the specialist side, and Maastricht and Groningen are more on the generalist side.

After this first overview, we explore each approach separately as a thesis or antithesis in the coming sections and then discuss a possible synthesis, namely the personalized approach, both in dementia research and dementia care practice.

### 3.2. First Pattern: A Generalist Approach to Dementia

According to Table 2, the first pattern includes respondents who have a generalist vision in the fields of both research and practice (8/44; 18%) and those who hold a generalist vision in one field and a combined vision in the other field (6/44; 13.6%).

Professors from the ERGO cohort study in Rotterdam take a very general view from the perspective of public health: 

“*Let’s start with the generalist perspective first and then see what remains. If all the smoking, all that hypertension, all that sugar, salt and alcohol were banned from society, that would reduce dementia by a third, just like many other lifestyle diseases! Put all your money in public campaigns and you will achieve your goal. That is not that sexy, but as a society we benefit the most*.”[R1]

Various professors argue in favor of breaking through the box thinking of syndromes and even warn that you would miss things and insights if you only think and act from the dementia image.

“*Older persons with dementia very often have multimorbidity and then the disease-specific approach is even a little absurd. With multimorbidity you have to talk about an integrated approach, I am very clear about that*.”[G5]

Although dementia requires subtle knowledge and competences about behavior, communication, and approach to maintain peace and space in the client system, there are many disadvantages of all kinds of disease-specific care providers, each of whom tackles a piece of the comorbidity. Furthermore, dementia is, in the first place, a fairly broad collective term, it is a syndrome. Secondly, there are a lot of co-morbidity and co-problems: 

“*It is always about interplay, in the particular context of a person. So, there are a lot of syndromes, in which there is a specific component and a more general other side. I think that a lot of mechanisms and forms of care and treatment can help more categories of people*.”[A11]

In the Dutch National Program for the Elderly, in networks around the eight academic centers in the period 2008–2017, it was experienced that the frailty of the elderly is not dependent on diagnosis and that for the elderly the focus on the quality of life and daily social functioning is more important than the underlying diseases. One of the leading senior professors explains: 

“*This focus also means that a more generalist or holistic approach often works better for elderly people living at home than a specialist approach that puts the disease first. Elderly people with dementia also have a lot of other things. Those people just have to be checked regularly on everything, because there is often more to it. That’s just basic prevention for everyone. Also, for dementia. As long as we don’t do that well in a generalist approach, we shouldn’t start on a more disease-specific approach either*.”[G2]

A recently appointed professor in nursing sciences is clear about a generalist profile: 

“*I attach great value to a generalist approach when it comes to the role of (district) nurses. There are many disease-specific professions, links, work approaches, et cetera. I think the case manager dementia is a very good example of this. It is filled in differently at each location and it is not possible to get it off the ground and demonstrate the added value. For nursing, I think it should be generic with good competences, which you can apply also to people with dementia*.”[M6]

A neurologist professor takes a generalist view on brain diseases: 

“*I just want to understand how the brain works. And then you also understand how memory is made and what can be destroyed when you have a particular type or mixed form of dementia. The processes underneath are important for almost all brain diseases*.”[U2]

The same professors from ERGO stress the generalist epidemiological view further: 

*Nowadays, in cohort studies several neurological disorders are taking together to create a whole of ‘major neurological diseases’. This requires an even more extreme generalist approach, because we know that there are many overlapping mechanistic processes. If you take stroke, Parkinson’s and dementia together, you can gain new insights.*”[R1]

An Amsterdam professor with anthropological background is in favor of a more universal human approach: 

“*Universal human needs are put under pressure by the disruptions of a disease. For example, your role as a partner or the daily things you do in life that come under pressure. That’s what bothers people most. It is not only about the disease, but it is also—and perhaps above all—about the consequences of the disease; the impact on ordinary life*.”[A17]

Summarizing the arguments for generalist orientations (see also Figure 2), dementia professors stress the holistic image of human beings, putting the person and quality of life central instead of the disease and impairments. Further arguments come from public health and epidemiological arguments (interferences with multimorbidity in aging populations), but also from systems and organisation perspectives, to organise and finance dementia care in more general instead of disease-specific mechanisms to develop sustainable health care systems and scarce labor forces. 

### 3.3. Second Pattern: Combinations of Generalists and Specialists Approaches

Table 2 shows that combined generalist-specialist visions in both research and practice are the most common (14/44; 31,8%). Dementia and its impact on the patient and family system require subtle knowledge and experience, so generalists in primary care should be well assisted by experts such as dementia case managers and geriatric medicine specialists, according to several dementia professors. 

“*If you take a disease-specific approach, you naturally run the risk to oversee other diseases and the patient as a person. And vice versa, if you make it all too generalist, then you run the risk that you simply do not see or do not tackle very basic things about dementia. So basically, you need a nice combination to understand the complete picture and really see the whole person*.”[T1]

An Amsterdam professor on psychosociological interventions seems to be on the same line: 

“*All people with chronic diseases have to deal with it in one way or another and that is actually a general model. I have worked it out for people with dementia to clarify what problems people with dementia encounter and that is specialized. We could learn a lot more from each other from the generalist perspective, but you need people with specific knowledge to guide people with dementia and their informal carers. So I am in favor of a more generalist approach, but with knowledge of the specific problems of the different patient groups. But generalist knowledge should also be present. Otherwise, there may be a danger that you reinvent the wheel.”*[A1]

A second Tilburg professor takes an organizational perspective: 

“*So maybe there is some kind of generalist basis on the level of how to organize it. But the closer you get to the individual client with dementia, the more specialist knowledge is needed*.”[T2]

A former-GP professor prefers a strong first-line team in a kind of shop around the GP: 

“*As a general practitioner you can arrange that care very well, but you should have some kind of shop. For example, that you can fly them in an elderly psychiatrist or geriatric medical specialist the moment you need them. Or that you can offer a case manager dementia, and that it is available immediately. After all, a general practitioner sees very little dementia and the specialist in elderly care a lot. So it is important that that expertise is available*.”[L2]

Another respondent includes arguments of efficiency and finance: 

“*Involving the right people at the right time, of course, remains the challenge we face. Because you cannot put all those specialists around the person with dementia from the start. Of course, that also makes it unnecessarily expensive*.”[A12]

From a policy and organizational perspective, there is also a plea for a layered approach: healthy aging should be the foundation, with special attention to the correct support for and interaction with people with dementia and their caregivers, because you do need specific knowledge and skills. Such an approach to a dementia-friendly society also works well for other vulnerable groups of citizens at the same time. A geriatric professor at Rotterdam uses the English ABC (Activities, Behavior, Cognition) model in his consultation room: 

“*Which Cognition problems arise in dementia, how can this influence someone’s Activities and Behavior now and in the future, and then discuss what can be done now and later. Such a focus on functioning, from insight and understanding, works on an individual and group level, both in aging in general and in dementia in particular*.” [R2]

A recently appointed professor in Nijmegen also emphasizes the generalist principles from the professional perspective of occupational therapy in combination with the disease-specific interpretation that she gives in the second instance of dementia. 

“*Yes, my answer is actually both. And I will explain, because from my background in occupational therapy I am actually of the generalist approach and I am also a health scientist. You have to look at the person, who is this person? Then, how can you ensure that this person, with all his possibilities and limitations, can do all those things that are meaningful to him. And how does the underlying disease influence that? So we start from the human being, in his social context. If you know our methodical approach, you can apply it to many target groups. However, you cannot unleash someone without disease-specific knowledge on people with dementia*.”[N1]

The required combination of generalist and specialist knowledge may change during the progression of dementia and along the care continuum from primary care to 24/7 intensive care. At the end of the care continuum, the Netherlands has developed a strong tradition of psychogeriatric nursing homes. 

“*For dementia as a total syndrome, a separate approach is absolutely necessary, because the disease affects you at the core of your existence. Nursing homes are completely tuned to this. They have a dementia-friendly society that you will not get in a residential complex with only a few people with dementia. But within the total syndrome, specific approaches to specific etiologies do not work, because the care has to look at what is bothering someone and how you can improve the quality of life. That is much more important than any etiological mark on it*.”[U1]

Overlooking the arguments for a mixed generalist-specialist orientation on dementia (see also Figure 2), many professors stress the benefits of interdisciplinarity, both in research and care practice. In care delivery there are strong advocates for person-centered and integrated care in a stepped care continuum, going back and forth between general primary care to specialized tertiary care. One professor formulated an interesting paradox that requires further thinking: to take generalist/normalizing approaches we need very specific knowledge about dementia diseases.

### 3.4. Third Pattern: Specialized Approaches to Dementia

According to Table 2, the last pattern includes respondents who have a specialist vision for both the field of both research and practice (9/44; 20.5%) and those who hold a specialist view on practice and combined vision on research (4/44; 9.1%). 

An emeritus professor has a clear view of the need for a specialist approach: 

“*I think dementia is so complex and so complicated that it requires a specific approach. Where that does not happen, for example in a hospital, people do not know what to do with it. It is an art to be able to deal well with people with dementia, and fortunately we now have quite a bit of knowledge about it. Fundamental knowledge about brain damage and that kind of work does really help. Then you can blame the disease rather than the human being. And you can also better understand and handle behavioral issues*.”[N5]

A professor in elderly psychiatry further elaborates on the behavioral aspects: 

*“In the course of the dementia, almost everyone has to deal with behavioral problems and you see that every team working with dementia is experienced at some point. And if you then scale up to a different team or setting, that is only disruptive for people and causes further serious behavioral problems. So, I would much rather have it in one hand or one care path. Also in terms of expertise that people simply know all phases of dementia well and not in silo’s who does what in what phase*.”[G3]

The onset of the disease, the growing cognitive impairments, and the dynamic impact on behavioral changes in dementia are so complex and time-consuming that care and support at home often result in a lot of misunderstanding, worry, and hassle. A director-professor of an Alzheimer Center makes a strong comparison: 

“*Dementia can be compared to metastasis in cancer, once you need care, it has to be specialized. Therefore, the Alzheimer Centers have a strong favor of specialized case management for dementia*.”[A13]

“*If you want to advance science, it makes sense to look at dementia separately. In research you want to have patient populations that are as clearly defined as possible*.”[A15]

A colleague at the same Alzheimer’s Center explains the usefulness of specialized case management dementia: 

“*The arrival of case managers has really helped and provided a lot of peace of mind. I have also noticed that here very much with one of our nurse specialists. He says: my work has simply changed. Because when we started here, I had a lot of work to do to arrange care all over the country. Which of course is not possible. But that is no longer necessary, because most people come in with a case manager so then they give back to that case manager afterwards. So that was a huge improvement. But recently, it’s falling apart. That is a great loss, I think it is a great shame to let go of case management for dementia. Also because I have the idea that, for example, general practitioners are not at all interested in a role of coordinating a dementia patient*.”[A2]

In non-Alzheimer’s dementias, such as FTD, the disease and its consequences are considered to be so specific that case management is often recognized as useful and necessary to optimize the system. 

“*Optimizing the system for a younger person is completely different than for an older person. It’s a much more complex system including work, participation issues, giving meaning, family issues and the like. That requires specific knowledge and guidance*.”[R3]

Even a convinced generalist in the wider field of elderly care endorses the exception for FTD: 

“*Even if you are a general practitioner who works perfectly with a district nurse, who invests energy in the different districts, who knows the districts and the social teams well, you could still very quickly get stuck with young people with dementia. Because GPs don’t see that much and have not enough experience. FTD requires specialized knowledge and signaling and communication skills. You must be able to read and attack very subtle signals*.”[M2]

The benefits of a disease-specific approach are further explained by a recently appointed professor of geriatrics and gerontology: 

“*The approach to people with dementia must be specific. Not so much the expertise, because I do believe that a district nurse with years of experience has the necessary knowledge. In our hospital, nurses are geriatrics, but some have a lot older patients with dementia and others only rarely. They often know exactly how to deal with such a patient, for example, to ensure that medication is taken in properly. We should not focus so much on technical specialization, but dementia care revolves around the little things in communication, approach and behavior, with your focus always on functioning. So the approach must be specific*!”[R2]

To summarize, the main argument for specialist approaches is in developing and implementing sound knowledge infrastructures in both research and care practice (see Figure 2). There are still big mysteries about the etiologies and disease mechanisms in various dementias and brain mechanisms. Coping with dementia requires subtle knowledge of how people express themselves, about practical, behavioral, and relationship problems in daily life, and about evidence-based interventions, in medicine, treatment, psychosocial support, case management et cetera. With the aging of populations comes a huge challenge to adequately support the doubling numbers of persons with dementia in all domains of human life by knowledgeable labor forces. 

## 4. Discussion

### 4.1. Developing a Synthesis 

Person-centered, integrated, and evidence-based dementia care requires strong connections between research, policy, and practice, in a delicate balance between generalist-specialist knowledge domains. The main arguments in the generalist-specialist debate are summarized in Figure 2.

Building on these insights, a new synthesis unfolds between the thesis of generalist and the antithesis of specialist care. Namely: personalized dementia care, aimed at the individual in his own micro-system. The terms “personalized”, “precision”, “stratified”, and “P4” (predictive, preventive, personalized, and participatory) medicine are used interchangeably to describe this concept, though some authors and organizations use these expressions separately to indicate particular nuances [24]. One professor formulates the synthesis nicely: 

“*You need very specific knowledge about the disease to take a generalist or normalizing approach. That’s actually a bit paradoxical*.” [M4]

More researchers and practitioners are now focusing on the capacities and possibilities of every person with dementia, instead of illness and disabilities. This requires broader interdisciplinary frameworks of interpretation, with a more social orientation that focuses on the impact on the personal environment of people with dementia and their loved ones. This is a plea for a holistic approach of each individual in his own unique environmental context, with a focus on daily and social functioning (social health), both in research and practice. A further challenge is to see this from the perspective of context or environment and the system world, which must always facilitate and stimulate this. The development of a dementia-friendly community or a dementia-inclusive society [25], and, in particular, the psychosocial interventions on which various professors from Nijmegen, Amsterdam, and Maastricht have been working for years, also fit into this approach. 

Due to the increasing economic and social impacts associated with aging populations, there is increasing health policy pressure, worldwide, to shift care away from specialist secondary care to generalist primary care [24]. In primary care, improving dementia care is best considered within a generalist approach across chronic conditions, since people living with dementia typically have multiple conditions to be managed, but primary care nurses find asking about cognitive decline difficult, due to a lack of knowledge and experience, feelings of helplessness and negative attitudes about people living with dementia which can lead to an unwillingness to get involved [26]. Further work is needed to assess how to organize healthcare systems so that care provided by generalists and specialists is well coordinated, results in superior outcomes, and is evidence-based and cost-effective [12]. Further studies are needed to delineate the activities for which generalists and specialists should be responsible, in order to provide the highest quality of research and care while most effectively utilizing manpower and other resources. Finding new combinations of generalist and specialist orientations will foster the strive for integrated care, defined by the WHO as ‘services that are managed and delivered so that people receive a continuum of health promotion, disease prevention, diagnosis, treatment, disease-management, rehabilitation, and palliative care services, coordinated across the different levels and sites of care within and beyond the health sector, and according to their needs throughout the life course [27]. Accordingly, integrated care strategies can target different levels of service provision: clinical (micro) level, service or organizational (meso) level, or system (macro) level. Integration of health and social care is widely advocated as a way to improve person-centered and system-centered outcomes for the increasing numbers of people with dementia and comorbidity, with varying and sometimes complex health needs. While the world is searching for a cure or disease-modifying treatments, research on dementia care has shifted from a predominant focus on improving or maintaining cognition or decreasing behavioral changes to outcomes such as a better quality of life, positive living with dementia, reablement [28], compensation for disability and improving daily function, mood, social health, community participation and social and emotional communication [9].

Building multidisciplinary workforce capacity to better deliver integrated care models and meet the needs of older people is a key recommendation of the WHO World Report on Aging and Health and is consistent with emerging evidence for delivering integrated care for older people with complex health needs [29]. The systematic review of Schot et al. revealed that contributing to interprofessional collaboration is multifaceted [30]. They named three inductive categories of how professionals contribute to working together to resemble existing theoretical perspectives on professional work outside of the interprofessional healthcare literature. First, bridging gaps, with close connotations with the concept of boundary spanning. Second, negotiating overlaps in roles and tasks, from the perspective of healthcare delivery as a negotiated order. This theoretical perspective usually focuses on the professional power struggles in which professionals use their cultural, social, or symbolic capital in order to maintain or improve their own position, but Schot et al. highlight the possibility of “re-adjusting” roles and responsibilities if needed [30]. Third, creating spaces for collaboration is closely related to what Noordegraaf calls “organizing” interprofessional collaboration [31].

### 4.2. Limitations

To the best of our knowledge, this is the first time that all academic professors in the field of research and care practice have been interviewed at a national level and challenged to explicate their vision about their generalist-specialist orientations on dementia. We found none but one analogy in other diagnostic fields, with palliative care as the only exception [16,17]. Without a firm scientific base or embedding in literature, an explorative qualitative design with semi-structured interviews was necessary to gather first insights from these highly profiled dementia professors as academic leaders in their fields. We included almost all dementia professors in the Netherlands, with only a few non-included persons who themselves stated that dementia was not only a very small part (less than 10 to 15%) of their work. Besides this quantitative form of saturation (including all possible respondents), this study also seems to have reached qualitative saturation as the degree of new information declined to almost zero during the interview process. 

Only one researcher did all the work of recruitment of respondents, interviewing and data collection, the whole analysis (using all necessary methods of coding in ATLAS.ti and member checks for quotes from respondents who all signed informed consent), and crafting of this paper. The argument for this lies in the benefits of oversight of both the fields of research and practice and full and easy excess to the community of academic professors of which the author is a longstanding member (more than 25 years). As a method of peer debriefing, an earlier Dutch version of the interview results (as a chapter in an e-book) had been thoroughly read and commented on by two experts at the national level (see Acknowledgements) [21]. However, a focus group with the same respondents with the individual interview results as input could have strengthened this research. As a lighter alternative, the author and his sponsors (see Acknowledgements) organized a meeting on 14 November 2019 with the Minister of Health, Welfare, and Sports in The Hague and 20 of the 44 dementia professors. There, the results of this research have been underwritten and embraced as an inspiring invitation to further develop interdisciplinary programming of research and practice in the whole dementia field. In Autumn 2022, and in line with this, ZonMw granted five new interdisciplinary consortia for dementia research, into (1) the disease mechanisms behind Alzheimer, FTD, and vascular dementia, (2) lifestyle interventions, (3) risk- and protective factor, (4) diagnostics and prognostics, and (5) young onset dementias [32].

Another possible limitation is that the research was completed in one single country, with specific systems in both health and dementia care and its own scientific structures with many diverse universities (research, technical, applied, and privately funded universities for business and theology). Replication in other countries is advised and should be very helpful to strengthen the growing international multi- and interdisciplinary consortia working on the EU Joint Programma Neurodegenerative Disease (JPND) Research and WHO’s global plans for research and practice, recognizing dementia as a public health priority in all aspects and domains.

The interviews with the dementia professors made it clear that they don’t know and meet each other enough to develop close collaboration. The current competitive model for research funding between universities and even between faculties or departments within the same university has led to compartmentalization and perverse incentives. The priority now is publishing, promoting, acquiring, specializing, and profiling in order to build own’s career path and strengthen the position of his/her team. Breaking down walls and silos is the major challenge for a future-proof strategy in research and practice throughout the whole dementia field.

## 5. Conclusions

There are varying arguments to hold a generalist and/or specialist vision and approach to dementia, both in research and care practice. This paper reveals that Dutch academic professors (N = 44) in all fields of dementia care and research have very mixed views on this balance. One-third pleas for a specialist orientation, one-third for a generalist approach, and one-third chooses a mix between generalist and specialist orientations, with some minor accents when differentiating between care and research. Each group has arguments for its generalist-specialist vision, but the synthesis suggests a paradigm of personalized and integrated dementia care. The task in dementia care is to keep the growing number of persons with dementia central as fully-fledged human beings, living at their homes in their communities as long and as well as possible. This requires combined generalist and specialist approaches with interdisciplinary frameworks to really arrive at person-oriented diagnostics and treatment, and more holistic care and support interventions, both in dementia research and care practice.

## Figures and Tables

**Figure 1 ijerph-20-03835-f001:**
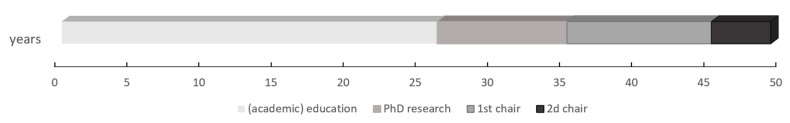
The average lifeline to become a dementia professor in the Netherlands (in years).

**Figure 2 ijerph-20-03835-f002:**
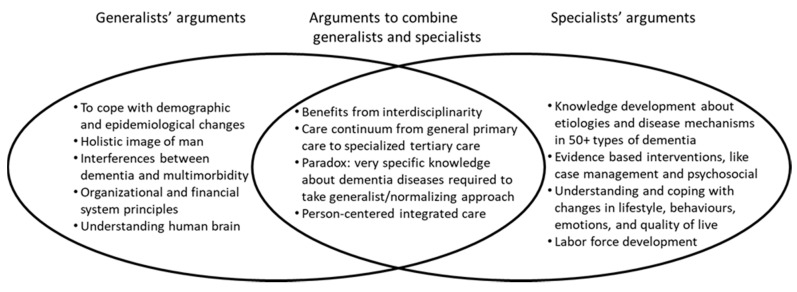
Main arguments for generalist-specialist approaches in the dementia field.

**Table 1 ijerph-20-03835-t001:** Characteristics of dementia professors in the Netherlands (N = 44; 2019).

Present university		
- Amsterdam UMC [A1-A17]	17	38%
- Groningen [G1-G7]	7	16%
- Maastricht [M1-M5]	5	11%
- Nijmegen (N1-N5]	5	11%
- Rotterdam [R1-R4]	4	9%
- Leiden [L1-L2]	2	5%
- Tilburg [T1-T2]	2	5%
- Utrecht [U1-U2]	2	5%
Gender		
- Female	19	43%
- Male	25	57%
Age: average	54.2 (s.d. 9.5)	
- <39 years	1	2%
- 40–44 years	6	14%
- 45–49 years	8	18%
- 50–54 years	11	25%
- 55–59 years	3	7%
- 60–64 years	10	23%
- >65 years	5	11%
Original disciplines		
- Medicine	24	55%
- Psychology	7	16%
- Health sciences	4	9%
- Social sciences	4	9%
- Biology	3	7%
- Other	2	5%
Present disciplines		
- Medicine	4	9%
- Geriatric medicine	7	16%
- Geriatrics	3	7%
- Psychiatry	3	7%
- Psychology	7	16%
- Neurology	5	11%
- Biology	3	7%
- Nurse sciences	3	7%
- Social sciences	3	7%
- Other	6	14%
Years in first chair: average	9.3 (s.d. 7.9)	
- <5 years	13	30%
- 5–9 years	14	32%
- 10–14 years	8	18%
- 15–19 years	6	14%
- >20 years	3	7%

**Table 2 ijerph-20-03835-t002:** Distribution in the generalist-specialist debate in research and practice (N = 44).

Approach in Dementia Research:	Approach in the Practice of Dementia Care:			Total
	Generalist	Combined	Specialist	
Generalist	8	4	3	15
Combined	2	14	3	19
Specialist	1	0	9	10
Total	11	18	15	44

Pearson Chi-square = 29.042; *p* < 0.001.

## Data Availability

Interview transcriptions are available from the author.

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
