# Peer review of "Understanding and Balancing Generalist-Specialist Approaches in Dementia Research and Care Practice, Qualitative Research with 44 Dementia Professors in The Netherlands"

_ijerph, 2023, doi:10.3390/ijerph20053835_

Round 1

Reviewer 1 Report

The topic is of importance to the field. It would be good if some points in the discussion part is summarized into visuals for better reading and understanding.

Author Response

  • Reviewer’s Comment: The topic is of importance to the field. It would be good if some points in the discussion part is summarized into visuals for better reading and understanding.
    Author’s Response: Thank you for stressing the importance to the field. Your suggestion to add visuals in the Discussion was a challenge for me. Hopefully you find the developed figure 2 in in the Discussion helpful.

Reviewer 2 Report

The manuscript entitled “Understanding and balancing generalist-specialist approaches 2 in dementia research and care practice, qualitative research 3 with 44 professors in the Netherlands” is interesting. However, there are many statistics reported in introduction come without citations. Also, there are some claims that is not tally, i.e., number of respondents (title claim 44 respondents, while on line 197 & line 150, author claimed 45 respondents). There are some concerns that shall be addressed before could be consider for publication.

1.      Line 30- Authors ask the reader to refer factsheets WHO. However, it is not attached as additional attachment/ link/ citation is not provided right after the statement.

2.      Line 38 & 39- Citation is needed for the statistics. Please check all throughout the manuscript.

3.      Line 158- “This paper’s aim is to unravel the arguments in the discussion about generalist xxxxxxx”. I think this statement is not appropriate. Less than 50 samples from same country seem like too limited for research to have such a claim. It is more to “serve as an additional finding to support xxxx”.

4.      Past study has reported that age gap could contribute to different opinion. Highly suggest the authors to discuss on this as well.

5.      Sub sections 3.1- 3.3 shall be revised. The current form is not organised/ systematically presented. A good manuscript shall deliver the idea clearly to the audience. Could consider use table to tabulate and present the info collected from the questionnaire.

6.      Conclusion shall be under a separate section. Please revise and extract partly of the discussion into conclusion.

Author Response

  • Reviewer’s Comment: The manuscript entitled “Understanding and balancing generalist-specialist approaches 2 in dementia research and care practice, qualitative research 3 with 44 professors in the Netherlands” is interesting.
    Author’s Response: Thank you for your comment that the manuscript is interesting.
  • Reviewer’s Comment: However, there are many statistics reported in introduction come without citations. Also, there are some claims that is not tally, i.e., number of respondents (title claim 44 respondents, while on line 197 & line 150, author claimed 45 respondents).
    Author’s Response: I removed myself from the sample and changed everything in text, tables and figures to 44 respondents.

Reviewer’s Comment: There are some concerns that shall be addressed before could be consider for publication.

  1. Reviewer’s Comment: Line 30- Authors ask the reader to refer factsheets WHO. However, it is not attached as additional attachment/ link/ citation is not provided right after the statement.
    Author’s Response: Thank you for signaling this omission. As reference [1], I included a hyperlink to the WHO factsheets.
  2. Reviewer’s Comment: Line 38 & 39- Citation is needed for the statistics. Please check all throughout the manuscript.
    Author’s Response: I included the WHO-reference [1] again in line 40.
  3. Reviewer’s Comment: Line 158- “This paper’s aim is to unravel the arguments in the discussion about generalist xxxxxxx”. I think this statement is not appropriate. Less than 50 samples from same country seem like too limited for research to have such a claim. It is more to “serve as an additional finding to support xxxx”.
    Author’s Response: Thank you for this reflection. On the one hand, all dementia professors working in the Netherlands during the research period, have been included. On the other hand, the reviewer is right that the whole sample is just from one single country, although heavily involved in many international consortia like those in the EU Joint Programma Neurodegenerative Disease Research (reference [3]). Lines 158-159 are adjusted into: “This paper’s aim is to explore the arguments in the discussion about generalist versus dis-ease-specific approaches to dementia in the Netherlands.” Also adjusted in the Abstract and Discussion.
  4. Reviewer’s Comment: Past study has reported that age gap could contribute to different opinion. Highly suggest the authors to discuss on this as well.
    Author’s Response: Thank you for your observation. Suggestions for relevant references to be included would have been very welcome. The distribution between generalist and specialist views in Table 2 had no significant age differences; included in the text at line 149.
  5. Reviewer’s Comment: Subsections 3.1- 3.3 shall be revised. The current form is not organized/ systematically presented. A good manuscript shall deliver the idea clearly to the audience. Could consider use table to tabulate and present the info collected from the questionnaire.
    Author’s Response: Thank you for your advise to better organize the subsections 3.1-3.4. Section 3.1 and particularly Table 2 provide the organization of different patterns in the views of the professors. To stress this, the titles of the subsections have been adjusted and each section starts with a connection with Table 2. Section 3.1 is restructured to strengthen the line of first results.
    Also subsections 3.2 and 3.3 have been restructured and sometimes a less relevant quote has been removed to keep a focus on the story line. The new figure 2 may be helpful as overview of the main arguments in the generalist and/or specialist debate.
  6. Reviewer’s Comment: Conclusion shall be under a separate section. Please revise and extract partly of the discussion into conclusion.
    Author’s Response: Thank you for your very justifiably I added a separate section 5. Conclusion and replaced some elements from the Discussion to the Conclusion.

Reviewer 3 Report

The results are relevant and unpublished, but some aspects of the methodology need to be clarified.

My considerations are about the methods, based on the Consolidated criteria for reporting qualitative research (COREQ).

In item 2.1 Design, clarify which methodological orientation was declared to support the study? For example: grounded theory, discourse analysis, ethnography, phenomenology and content analysis. In item 2.2. Interviews, clarify the characteristics of the interviewers and what the participants knew about themselves or about the main researcher.

Regarding the analysis methodology, clarify: what methodological orientation was used, how many data coders were used, whether themes were identified in advance or derived from the data, and whether participants provided feedback on the results.

Indicate the submission and approval of the research project to a research ethics committee.

Author Response

  • Reviewer’s Comment: The results are relevant and unpublished, but some aspects of the methodology need to be clarified.
    Author’s Response: Thanks for your support for this paper and your efforts to advise on improvements.
  • Reviewer’s Comment: My considerations are about the methods, based on the Consolidated criteria for reporting qualitative research (COREQ). In item 2.1 Design, clarify which methodological orientation was declared to support the study? For example: grounded theory, discourse analysis, ethnography, phenomenology and content analysis. In item 2.2. Interviews, clarify the characteristics of the interviewers and what the participants knew about themselves or about the main researcher.
    Author’s Response: I included a starting sentence in subsection 2.2 to clarify that all work has been done by one single person; this was already discussed in subsection 4 in the first version of this manuscript. A new section 2.3 is included to clarify the methodological orientation (in line with grounded theory). This section also included two new references [22] and [23], the remainder after 23 is renumbered with track changes in the text and references.
  • Reviewer’s Comment: Regarding the analysis methodology, clarify: what methodological orientation was used, how many data coders were used, whether themes were identified in advance or derived from the data, and whether participants provided feedback on the results.
    Author’s Response: As mentioned above and added tot section 2.2, the author did everything alone and thus was the single data coder. The last part of this section already explained the member check with respondents, I hope that this way of organizing the feedback of participants is satisfactory.
  • Reviewer’s Comment: Indicate the submission and approval of the research project to a research ethics committee.
    Author’s Response: Thanks for your comments, which I also discussed with the Associate Editor. Recruiting was done by a personal email-invitation, including objectives and way of interviewing. Informed consent given by return-email and was also the first question in the interview itself, written down in the full transcript of the interview and send to respondent for approval (‘member check’) which I received from everybody by return-email.
    These peer-to-peer interviews about professional work did not use in any way patient data. The demographic data about the professors themselves come from the approved CV's of each person, compiled from public data on university sites, approved by the respondents themselves and published in Dutch e-book (reference [21]  in the manuscript).

Round 2

Reviewer 2 Report

The introduction has been sufficiently revised. However, as suggested in the first revision, the result section shall be heavily revised. Tabulating the feedback in table form would be easier to follow.

Author Response

I'm grateful to Reviewer #2 for reading and commenting the revised version again.
His comments are twofold:

1. Reviewer’s Comment: The introduction has been sufficiently revised.
    Author’s Response: Thank you for your appreciation.

2. Reviewer’s Comment: However, as suggested in the first revision, the result section shall be heavily revised. Tabulating the feedback in table form would be easier to follow.
Author’s Response: Thank you for your comment on the Results section. Obviously, my first revision (with new introductions to connect that particular subsection to Table 2 in the first subsection 3.1) did no hold up to your expectations, I’m sorry for that. However, I strongly feel that your suggestion to tabulate the results is not compatible with the qualitative design of this study. As an alternative, I tried to strengthen the three subsections 3.2-3.4 with a summarizing paragraph, also building up and connecting to the summarizing Figure 2 in the Discussion.
So I hope that these summaries in subsection 3.2 (yellow lines 390-395), subsection 3.3 (yellow lines 457-462) and subsection 3.4 (yellow lines 516-523) will help the reader to grasp the essence of the various arguments from dementia professors in this complex generalist-specialist debate, also presented in Figure 2 (which was already added in the first revision, but is now better embedded in the paper). To introduce the objective to facilitate the reader in his own thinking about the generalist-specialist debate on the basis of various detailed quotes, I also extended section 2.3 Data analysis (yellow lines 195-202).

Hopefully, the combination of this second revision with the first revision may now satisfy Reviewer #2.

Reviewer 3 Report

Dear authors

The answers are in accordance with the requests, I recommend the publication and congratulate you for the work.

Author Response

I'm grateful to Reviewer #3 for appreciating the way her/his comments have been answered in the revised version. Thanks for your recommendation to publish this paper, with congratulations for the work. Much obliged!